# Spatial organization shapes the turnover of a bacterial transcriptome

**Jeffrey R Moffitt[1,2], Shristi Pandey[3], Alistair N Boettiger[1,2], Siyuan Wang[1,2], Xiaowei Zhuang[1,2,4]***

[1]Howard Hughes Medical Institute, Harvard University, Cambridge, United States; [2]Department of Chemistry and Chemical Biology, Harvard University, Cambridge, United States; [3]Department of Molecular and Cellular Biology, Harvard University, Cambridge, United States; [4]Department of Physics, Harvard University, Cambridge, United States

**Abstract** Spatial organization of the transcriptome has emerged as a powerful means for regulating the post-transcriptional fate of RNA in eukaryotes; however, whether prokaryotes use RNA spatial organization as a mechanism for post-transcriptional regulation remains unclear. Here we used super-resolution microscopy to image the *E. coli* transcriptome and observed a genome-wide spatial organization of RNA: mRNAs encoding inner-membrane proteins are enriched at the membrane, whereas mRNAs encoding outer-membrane, cytoplasmic and periplasmic proteins are distributed throughout the cytoplasm. Membrane enrichment is caused by co-translational insertion of signal peptides recognized by the signal-recognition particle. Time-resolved RNA-sequencing revealed that degradation rates of inner-membrane-protein mRNAs are on average greater that those of the other mRNAs and that this selective destabilization of inner-membrane-protein mRNAs is abolished by dissociating the RNA degradosome from the membrane. Together, these results demonstrate that the bacterial transcriptome is spatially organized and suggest that this organization shapes the post-transcriptional dynamics of mRNAs.

**\*For correspondence:** zhuang@ chemistry.harvard.edu

## Introduction

In eukaryotic systems, the spatial organization of the transcriptome plays a fundamental role in regulating the post-transcriptional fate of RNA. Such organization leads to spatially localized translation and degradation of mRNAs, which are essential for a diverse set of biological behaviors including cell motility, cellular polarization, and stress response (*Balagopal and Parker, 2009*; *Buxbaum et al., 2015*; *Holt and Schuman, 2013*; *Martin and Ephrussi, 2009*). By contrast, spatial localization has not been considered to play a significant role in the post-transcriptional dynamics of bacterial mRNAs.

Early measurements of the dynamics of a handful of synthetic mRNAs within bacterial cells suggest that mRNAs are more or less uniformly distributed inside the cells. Single fluorescently labeled synthetic mRNAs have been observed to diffuse freely throughout the cytoplasm in *E. coli* (*Golding and Cox, 2004*; *2006*), and the local fluorescent signals from labeled mRNAs appear to fluctuate in time in a manner consistent with free diffusion (*Le et al., 2005*; *Valencia-Burton et al., 2009*). However, recent evidence has begun to reveal that some native mRNAs do not diffuse freely throughout the cell but are rather localized to specific cellular compartments (*Campos and Jacobs-Wagner, 2013*; *Nevo-Dinur et al., 2012*). Different spatial patterns of mRNAs have been identified. In one of the patterns, mRNAs have been observed to reside in the vicinity of the DNA loci from which they were transcribed. This pattern was observed for groESL, creS, divJ, ompA, and fljK transcripts in *C. crescentus* and the lacZ transcript in *E. coli* (*Montero Llopis et al., 2010*). Similarly, the

**eLife digest** Within a cell, molecules of messenger RNA (mRNA) encode the proteins that the cell needs to survive and thrive. The amount of mRNA within a cell therefore plays an important role in determining both the amount and types of proteins that a cell contains and, thus, the behavior of the cell.

In eukaryotic organisms, like humans, it has been established that it is not just the amount of mRNA that influences cell behavior, but also where the mRNA molecules are found within the cell. However, in bacteria, which are much smaller than human cells, it has long been believed that the location of an mRNA within the cell does not affect its behavior. Despite this, recent studies that have looked at small numbers of bacterial mRNAs have shown that some of these molecules are found in larger numbers than usual at certain sites inside cells. This suggests that location may actually affect the activity of some bacterial mRNAs. But do similar localization patterns occur for all of the thousands of different mRNAs that bacteria can make?

To address this question, Moffitt et al. developed an approach that allows large, defined sets of mRNAs to be imaged in bacteria. Using this approach to study *E. coli* revealed that a considerable fraction of all the mRNAs that these bacteria can make locate themselves at specific sites within a cell. For example, mRNAs that encode proteins that reside inside the cell's inner membrane are found enriched at this membrane. This localization also plays an important role in the life of these mRNAs, as they are degraded more quickly than those found elsewhere in the cell. This enhanced degradation rate arises partly because the enzymes that break down mRNA molecules are also found at the membrane.

Thus, bacteria can shape the process by which an mRNA is made into protein by controlling where in a cell the mRNA is located. The next steps are to understand why bacteria use cell location to influence the rate of mRNA degradation.

average distributions of the lacI mRNA in *E. coli* cells appear to show enrichment in a cellular region at which the lacI chromosome loci is also enriched (*Kuhlman and Cox, 2012*). A second, distinct pattern has been observed in which mRNAs do not reside near the DNA loci from which they are transcribed, but instead reside in the cellular compartment where their encoded proteins are localized. For example, the *E. coli* bglGFB, lacY, and ptsG mRNAs, which encode the inner-membrane proteins BglF, LacY, and PtsG, have been found enriched near the cell membrane (*Fei et al., 2015*; *Nevo-Dinur et al., 2011*); the bglG fragment of the *E. coli* bglGFB transcript and the *B. subtilis* comE transcript, which encode the polar localized BglG and ComEC proteins, respectively, have been observed enriched at the cell pole or at the septa of sporulating cells (*Nevo-Dinur et al., 2011*; *dos Santos et al., 2012*). In addition to these patterns, higher order mRNA structures have also be suggested, such as a helical RNA distribution near the cell membrane (*Valencia-Burton et al., 2009*). Because of the disparate spatial patterns that have been observed previously and the relatively small number of RNAs that have been investigated, it remains unclear how mRNAs are spatially organized inside bacterial cells and whether any of the observed spatial organizations is a genome-wide property or a special property of a small number of genes.

The molecular mechanisms responsible for mRNA localization in bacterial also remain incompletely understood. Several lines of biochemical evidence have revealed that mRNAs encoding inner-membrane proteins are, in part, translated at the membrane by the co-translational insertion of inner-membrane proteins into the membrane (*Driessen and Nouwen, 2008*). Thus, these mRNAs should spend at least a portion of their lifetimes near the cell membrane. However, co-translational insertion has yet to be linked to any of the reported RNA localization patterns. Instead, a recent study has suggested a translation-independent membrane localization mechanism for RNA based on the observation that inhibition of translation of the bglF transcript does not disrupt its membrane localization (*Nevo-Dinur et al., 2011*). This observation has led to the suggestion that an RNA zip-code in combination with unknown zip-code-binding proteins directs this bacterial RNA localization (*Nevo-Dinur et al., 2011*), similar to the established RNA localization mechanisms in eukaryotes (*Buxbaum et al., 2015*). However, the proteins responsible for identifying this putative zip-code

have not been described in bacteria nor has this localization mechanism been extended to any other bacterial mRNAs. Similarly, no mechanism is known for the retention of mRNA near the chromosomal locus from which they were transcribed.

Finally, it remains unknown what physiological consequences spatial organization might have on the post-transcriptional dynamics of mRNAs in bacterial cells. Interestingly, translation and RNA processing enzymes are not uniformly distributed in bacteria. For example, ribosomal proteins tend to be excluded from the nucleoid and enriched in the cell periphery and cell poles in both *E. coli* and *B. subtilis* (*Bakshi et al., 2015*; *Robinow and Kellenberger, 1994*). Core components of the RNA degradation machinery have been found enriched at the cell membrane in *E. coli* (*Mackie, 2012*) and *B. subtilis* (*Lehnik-Habrink et al., 2011*), and in the nucleoid in *C. crescentus* (*Montero Llopis et al., 2010*). Even components of the trans-translation pathway, a pathway responsible for the resolution of defective transcripts, appear to cycle between the cytoplasm and membrane as a function of cell cycle in *C. crescentus* (*Russell and Keiler, 2009*). Such non-uniform distributions of RNA-interacting proteins give rise to the possibility that spatial organization may play an important role in shaping the post-transcriptional dynamics of mRNAs, if the mRNAs are themselves not uniformly distributed. However, no evidence has been described yet for the role of spatial organization in the post-transcriptional fate of bacterial mRNAs.

Here we probe the presence, mechanism, and physiological consequences of the spatial organization of mRNAs in *E. coli* at the transcriptome scale. We developed a method to directly image the spatial organization of large but defined fractions of the transcriptome, and our measurements revealed transcriptome-scale spatial organizations of mRNAs that depended on the cellular locations of their targeted proteins: mRNAs encoding inner-membrane proteins were found enriched at the membrane whereas mRNAs encoding cytoplasmic, periplasmic and outer-membrane proteins were found relatively uniformly distributed throughout the cytoplasm. Genomic organization, on the other hand, did not appear to play a major role in the organization of the transcriptome in *E. coli*. We further demonstrated that co-translational insertion of signal peptides recognized by the signal-recognition-particle (SRP) was responsible for this membrane localization of inner-membrane-protein mRNAs. To explore the physiological consequences of this transcriptome-scale organization, we used time-resolved next-generation sequencing to measure mRNA lifetimes and found that the mRNAs encoding inner-membrane proteins were selectively destabilized compared to mRNAs encoding outer-membrane, cytoplasmic and periplasmic proteins. Finally, to elucidate potential mechanisms for this selective destabilization, we imaged the distribution of all of the enzymes associated with RNA processing in *E. coli* and observed that members of the RNA degradosome are enriched on the membrane. A genetic perturbation that removed these enzymes from the membrane preferentially stabilized mRNAs encoding inner-membrane proteins, suggesting that their physical proximity to the membrane-bound RNA degradosomes may be responsible for the native destabilization of these mRNAs.

## Results

### Spatial organization of the *E. coli* transcriptome depends on the location of the encoded proteins

To enable the direct measurement of the spatial distribution of mRNAs at the transcriptome scale, we developed a method that allows us to directly measure the spatial distribution of large but defined portions of the transcriptome. In particular, by selectively staining large mRNA groups which share common properties, we not only avoided inference of transcriptome-scale organization from measurements of only a few mRNAs but also directly tested the role of general mRNA properties on spatial organization (*Figure 1A*).

Our method is based on single-molecule fluorescence in-situ hybridization (FISH) (*Femino et al., 1998*; *Raj et al., 2008*). A challenge to simultaneously imaging large populations of mRNAs is the generation of complex FISH probe sets for labeling such RNA populations, each comprising hundreds to thousands of unique oligonucleotide probes. To overcome this challenge, we used an Oligopaint-based approach and took advantage of the ability of array-based synthesis to generate complex oligonucleotide libraries (*Beliveau et al., 2012*). Specifically, we designed libraries comprising sequences that target the desired RNAs flanked by primers that allow the selection, enzymatic

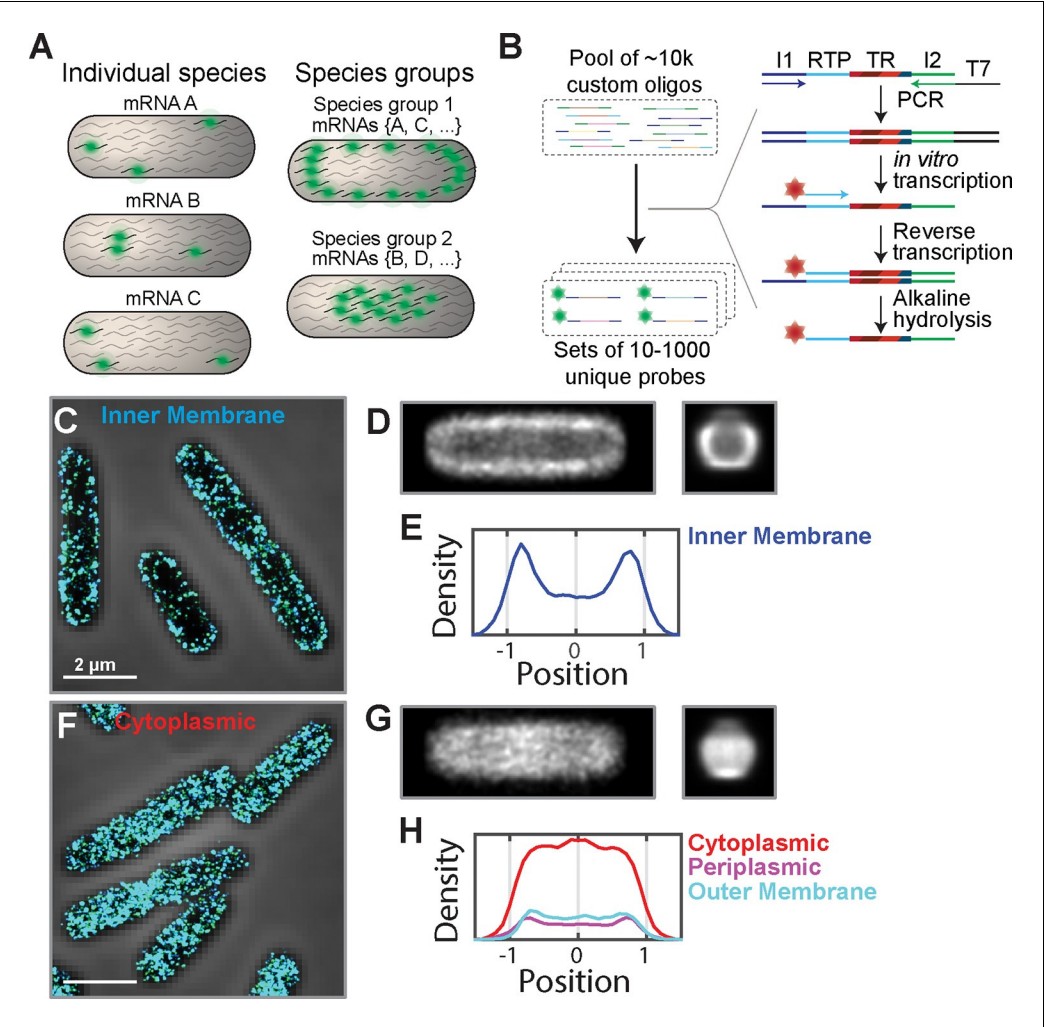

**Figure 1.** The *E. coli* transcriptome is spatially organized with inner-membrane-protein mRNAs enriched at the membrane. (**A**) A scheme illustrating fluorescent labeling and imaging of large but defined populations of mRNAs simultaneously instead of imaging one mRNA species at a time. (**B**) The required complex FISH probe sets are generated via enzymatic amplification of array-derived custom oligonucleotide pools containing tens of thousands of unique sequences. Subsets of these oligopools are selected via PCR, amplified and converted into RNA via in vitro transcription, converted back into DNA via reverse transcription with a fluorescently labeled primer. The RNA templates are removed by alkaline hydrolysis. I1 and I2 represent PCR primers unique for each probe set. RTP represents a reverse transcription primer common to all probe sets. TR (targeting region) represents the portion of the oligo complementary to one of the RNAs of interest. (**C**) Stacked phase contrast (gray) and STORM cross-section images (color) of example fixed *E. coli* cells stained with FISH probes against all mRNAs encoding inner-membrane proteins that are in the abundance range of 3–30 copies per cell. The STORM images of the middle section (300-nm thick) of cells are shown here. 3D-STORM images of the entire cells as well as images of mRNAs in other abundance ranges are shown in *Figure 1—figure supplement 1*. (**D**) Average short-axis (left) and long axis (right) cross-section images of inner-membrane-protein mRNAs derived from 611 cells computationally normalized to a common width and a common length and then aligned. (**E**) Density profile of inner-membrane-protein mRNAs constructed from the middle slice (150 nm) of the average long-axis cross-section image shown in (**D**, right). The x-axis is normalized to the radius of the cell. (**F, G**) Same as (**C, D**) but for mRNAs encoding cytoplasmic proteins in the abundance range of 3–30 per cell and the average cross-section images were derived from 319 individual cells. (**H**) Same as (**E**) but for mRNAs encoding cytoplasmic proteins (red), periplasmic proteins (purple), and outer-membrane-protein distributions (cyan). The cytoplasmic, periplasmic and outer-membrane-protein distributions were derived from 319, 338 and 194 cells, respectively. Scale bars: 2 μm.

The following figure supplements are available for figure 1:

*Figure 1 continued on next page*

*Figure 1 continued*

**Figure supplement 1.** Spatial organization of mRNAs that encode proteins residing in different cellular locations and are in different RNA abundance ranges.
**Figure supplement 2.** Being polycistronic with an inner-membrane-protein message can confer partial membrane enrichment to an mRNA.

amplification, and fluorescent tagging of defined subsets of these libraries, each of which target a specific group of mRNAs (*Figure 1B*). We then amplified these oligonucleotide templates to generate complex but defined sets of FISH probes by a high-throughput enzymatic amplification method (*Chen et al., 2015b*; *Murgha et al., 2014*).

We first used this approach to test whether the transcriptome of *E. coli* is organized based on the intracellular locations of the encoded proteins. To this end, we designed several FISH probe sets, each targeting a specific population of mRNAs whose encoded proteins reside within one of the four cellular compartments in *E. coli*: cytoplasm, inner membrane, periplasm, and outer membrane. To control for the large differences in mRNA abundance within a group, we further sub-divided each group by mRNA abundance. Within the final groups, no single mRNA species was predicted to produce more than ~10% of the signal from the imaged group. We fixed and labeled *E. coli* cells with these probes, and imaged the mRNA distributions using three-dimensional stochastic optical reconstruction microscopy (3D-STORM) (*Huang et al., 2008*; *Rust et al., 2006*).

The mRNA distributions showed a clear distinction between different mRNA groups: mRNAs that encode inner-membrane proteins were strongly enriched at the membrane (*Figure 1C–E* and *Figure 1—figure supplement 1A*) whereas mRNAs encoding cytoplasmic proteins were more or less uniformly distributed throughout the cytoplasm (*Figure 1F–H* and *Figure 1—figure supplement 1B*), except for some cases where we observed a moderate depletion of mRNAs from the nucleoid (*Figure 1—figure supplement 1B*). This difference is evident not only in the distributions of mRNAs in individual cells (*Figure 1C,F*) but also in the average mRNA distributions over several hundred imaged cells after normalization of the cell dimensions (*Figure 1D,E,G,H*). These different spatial distributions did not depend on the abundance range of the stained mRNAs (*Figure 1—figure supplement 1A,B*). Notably, mRNAs that encode periplasmic proteins and outer-membrane proteins, which reside within nanometers of inner-membrane proteins, did not show a strong enrichment at the membrane. Instead, these mRNAs were found distributed more or less throughout the cytosol (*Figure 1H* and *Figure 1—figure supplement 1C,D*), like those mRNAs that encode cytoplasmic proteins. Interestingly, among these latter groups, the subset of mRNAs that were polycistronic with inner-membrane-protein mRNAs also exhibited membrane enrichment (*Figure 1—figure supplement 2*), which explains the slight membrane enrichment observed in the mRNA populations encoding periplasm and outer-membrane proteins (*Figure 1—figure supplement 1C,D*). For all groups, the number of RNA localizations that we detected using mRNA-targeting probes was much larger (~10–100 fold) than that detected when using anti-sense probes with reverse compliment sequences (*Figure 1—figure supplement 1E,F*), indicating highly specific labeling.

## Spatial organization of the genome does not play a major role in the spatial organization of the *E. coli* transcriptome

The bacterial genome is spatially organized with defined genomic loci occupying defined locations within the cell as a function of the division cycle (*Wang et al., 2013*). To test whether this genomic organization plays a role in the spatial organization of the transcriptome, we constructed multiple FISH probe sets, each labeling the specific population of mRNAs that are transcribed from one of twenty different 100-kb chromosomal regions (*Figure 2*). Because such 100-kb regions occupy small volumes within the nucleoid (*Wang et al., 2013*), we would expect each mRNA probe set to produce one or a few bright fluorescent foci (one for each copy of the chromosome) within cells if these mRNAs resided near their corresponding DNA loci. Instead, the majority of these mRNA populations were uniformly distributed throughout the cytoplasm (*Figure 2*). Interestingly, a subset of these mRNA groups showed some membrane enrichment, and these groups were enriched for mRNAs that encode inner-membrane proteins.

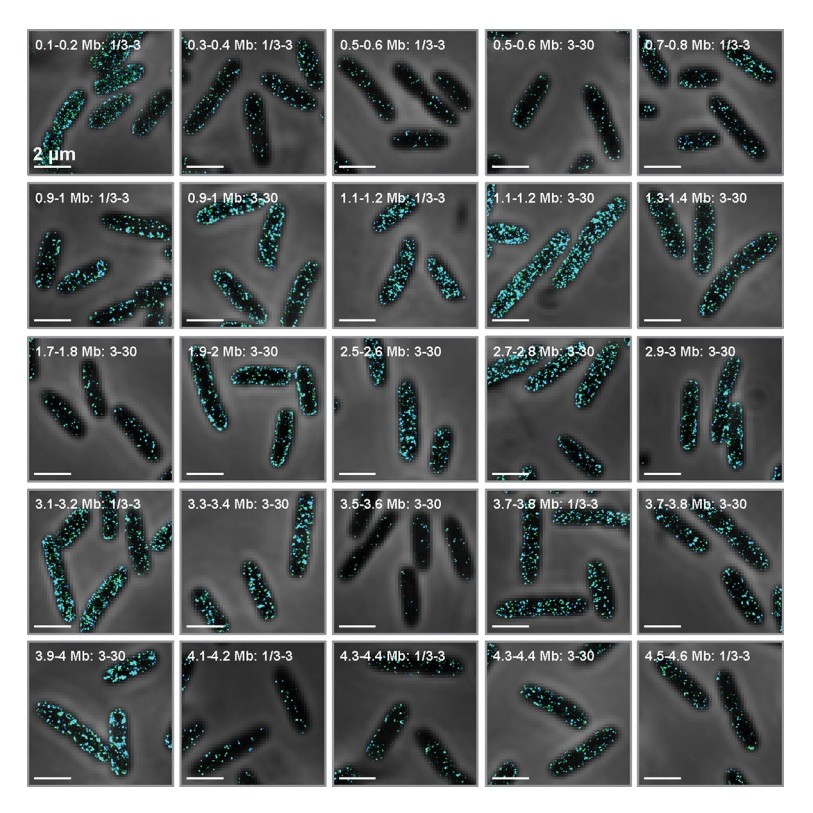

**Figure 2.** Genomic organization does not play a major role in the organization of the *E. coli* transcriptome. Stacked phase contrast (gray) and STORM cross-section images (color) for example fixed *E. coli* stained for all mRNAs transcribed from discrete 100-kb genomic loci in the abundance range of 1/3–3 and/or 3–30 copies per cell. The label marks the genomic region and abundance range studied in each case. Several cases show significant membrane enrichment, and these cases correspond to loci enriched in mRNAs that encode inner-membrane proteins. Scale bars: 2 μm.

The substantial fraction of the transcriptome probed in these measurements suggests the generality of our observed spatial patterns. In our study of the relationship between the spatial organization of the mRNAs and their encoded proteins, we stained 27% of all *E. coli* mRNAs and 76% of those actually expressed under the probed growth condition (>0.3 copies per cell). Similarly, in the study of the role of locations of the genomic loci on the spatial organization of their encoded mRNAs, we probed mRNAs transcribed from nearly half of the *E. coli* chromosome. Thus, we conclude that the patterns we observe here reflect the general behavior of *E. coli* mRNAs: mRNAs that encode inner-membrane proteins are enriched near the membrane whereas mRNAs that encode proteins that reside in the cytoplasm, periplasm and outer-membrane tend to be distributed throughout the cytoplasm, although we cannot rule out the possibility that there are exceptions to these general behaviors for some mRNAs. By contrast, the spatial organization of the genome does not appear to play a major role in shaping the spatial organization of the *E. coli* transcriptome.

## Co-translational insertion of membrane proteins is important for the membrane enrichment of the mRNAs encoding these proteins

Next we investigated the mechanism responsible for establishing this general spatial organization of the *E. coli* transcriptome. Given the correlation observed between the spatial organization of the mRNAs and their encoded proteins, we reasoned that the pathways involved in directing proteins to different cellular locations might be involved in establishing the spatial distribution of mRNAs. In bacteria, there are two major pathways responsible for protein targeting (*Driessen and Nouwen, 2008*): the signal recognition particle (SRP)-dependent pathway and the SecB-dependent pathway.

Most inner-membrane proteins use the SRP pathway, whereas most outer-membrane and periplasmic proteins use the SecB pathway. The SRP pathway is believed to co-translationally insert proteins into the membrane; therefore, mRNAs subject to this pathway would be translated, in part, at the membrane. By contrast, proteins destined for the SecB pathway are translated in the cytosol. Thus, the co-translational membrane insertion of the inner-membrane proteins via the SRP pathway would provide a simple explanation for the membrane enrichment observed for mRNAs encoding these proteins. However, this mechanism has not been linked to the previously observed mRNA distribution patterns and, for the one *E. coli* mRNA (bglF) whose membrane localization mechanism has been probed, it has instead been suggested that its membrane localization is translation independent and, thus, cannot be established by SRP-dependent co-translational insertion (*Nevo-Dinur et al., 2011*). Therefore, a critical test of the mRNA localization mechanism is needed.

The choice of SRP or SecB pathway is dictated by signal peptides near the N-terminus of the protein (*Driessen and Nouwen, 2008*). Thus, to test the role of co-translational insertion in the spatial localization of mRNAs, we created a series of fusion constructs between a test mRNA that encodes the fluorescent protein mMaple3 (*Wang et al., 2014*) and native signal-peptide sequences derived from different *E. coli* proteins (*Figure 3A*). We inserted these fusion genes into the chromosome and measured the spatial distribution of their mRNAs using FISH labeling and 3D-STORM imaging. Fusion to different signal-peptide sequences clearly directed the fusion mRNAs to different locations in the cell: mRNAs that were fused to the SRP signal sequences derived from the inner-membrane proteins FhuB, CcmH, and AcrB (*Huber et al., 2005*) were almost exclusively localized at the membrane (*Figure 3B–D*), whereas those mRNAs that were fused to the SecB signal sequences derived from the periplasmic proteins GlpQ, LivJ, PhoA, and MalE (*Huber et al., 2005*) were distributed throughout the cytoplasm (*Figure 3E,F*). As further evidence that the membrane localization of mRNAs is driven by the SRP targeting pathway, the signal-peptide sequence derived from TolB, one of the rare periplasmic proteins that uses the SRP pathway (*Huber et al., 2005*), also directed the mMaple3 mRNA to the membrane (*Figure 3C*). To determine if translation of the signal peptide is required for the membrane enrichment induced by the SRP signal sequences, we removed the start codon from the fusion mRNAs and found that this perturbation removed these mRNAs from the membrane (*Figure 3C,D*). Thus, translation of the N-terminal signal peptide that target proteins to the SRP pathway is required for directing these mRNAs to the membrane.

To test the translation-dependence of the membrane localization of mRNAs at the transcriptome scale, we used kasugamycin, a translation-initiation inhibitor, to release the cellular pool of mRNAs from ribosomes (*Schluenzen et al., 2006*) and re-measured the spatial distribution of the endogenous mRNAs encoding inner-membrane proteins. The membrane enrichment observed for these mRNAs was abolished by the kasugamycin treatment (*Figure 3G,H*), indicating that this transcriptome-scale pattern is translation dependent. Taken together, our results suggest that co-translational insertion of the inner-membrane proteins mediated by the SRP plays the major role in the membrane localization of mRNAs encoding these proteins.

Because our results contrast the translation-independent membrane-targeting mechanism previously proposed for bglF (*Nevo-Dinur et al., 2011*), we re-examined the localization mechanism of bglF. bglF is an inner-membrane protein and, as such, contains a SRP signal peptide in the N-terminal region of the protein (*Figure 3—figure supplement 1A*), and the nucleotide sequence encoding this signal peptide overlaps with the RNA region previously proposed to direct this transcript to the membrane via a translation-independent mechanism (*Nevo-Dinur et al., 2011*). To test the role of translation of this signal region on the localization of bglF mRNA, we created several fusion constructs between bglF derivatives and mMaple3 that are either translationally fully competent or translationally inhibited by insertion of stop codons before the SRP signal. Translationally competent bglF mRNAs were found enriched at the membrane (*Figure 3—figure supplement 1A–C*), as expected; however, constructs with stop-codon insertions that disrupted the translation of the SRP signal region, but which still contain the RNA sequence that encodes it, were no longer enriched at the membrane (*Figure 3—figure supplement 1D–I*). Thus, our results suggest that translation of the SRP signal sequence is required for bglF membrane localization, consistent with the transcriptome-wide mechanisms described above.

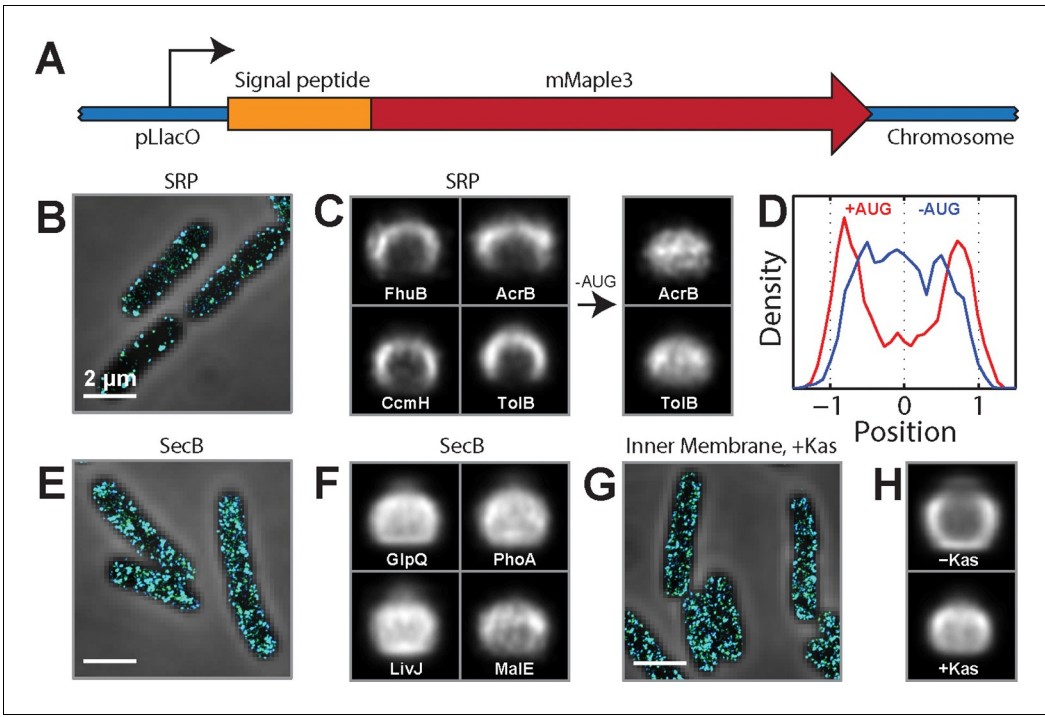

**Figure 3.** SRP-dependent co-translational insertion of signal peptides plays a major role in the membrane localization of inner-membrane-protein mRNAs. (**A**) Fusion constructs between different signal peptides and mMaple3. (**B**) Stacked phase contrast (gray) and STORM cross-section images (color) of example *E. coli* cells expressing mMaple3 fused to the signal peptide from an SRP-dependent protein FhuB. The cells were stained with FISH probes against mMaple3. (**C**) Left: Average long-axis cross-section images of cells expressing mMaple3 fused to SRP-dependent signal peptides derived from FhuB, CcmH, AcrB, and TolB. Right: Average long-axis cross-section images of cells expressing mMaple3 fusions to AcrB and TolB signal peptides without the start codon (-AUG). (**D**) Density profiles derived from the average long-axis cross-section images of mMaple3 fusions to the AcrB signal peptide with (red) and without (blue) the start codon. Density profile is as defined in *Figure 1E*. (**E**) Stacked phase contrast and STORM cross-section images of example *E. coli* cells expressing mMaple3 fused to a signal peptide derived from a SecB-dependent protein GlpQ. The cells were stained with FISH probes against mMaple3. (**F**) Average long-axis cross-section images of cells expressing mMaple3 fused to SecB-dependent signal peptides derived from GlpQ, LivJ, PhoA and MalE. (**G**) Stacked phase contrast and STORM cross-section images for example *E. coli* cells treated with the translation-initiation-inhibitor kasugamycin. The cells were stained with the FISH probe set against inner-membrane-protein mRNAs in the abundance range of 3–30 copies per cell. (**H**) Average long-axis cross-section images of cells in the presence (+Kas) and absence (-Kas) of kasugamycin. The cells were stained with the FISH probes against inner-membrane-protein mRNAs in the abundance range of 3–30 copies per cell. Average long-axis cross-section images in C, F and H were derived from all measured cells, tens to hundreds of cells in each case. Scale bars: 2 μm.

The following figure supplement is available for figure 3:

**Figure supplement 1.** mRNA for the inner-membrane protein BglF is enriched at the membrane in a translation-dependent fashion.

## Inner-membrane-protein mRNAs are preferentially destabilized

We next asked if this spatial organization has any physiological consequences on the post-transcriptional dynamics of *E. coli* mRNAs. To address this question, we used time-resolved RNA sequencing (*Chen et al., 2015a*; *Geisberg et al., 2014*; *Kristoffersen et al., 2012*; *Munchel et al., 2011*; *Rabani et al., 2011*) to simultaneously measure the degradation kinetics of all mRNA species in *E. coli* — a technique that we refer to as τ-seq hereafter. Briefly, we inhibited transcription initiation with the antibiotic rifampicin and then measured RNA abundance with RNA-seq at various time points after this treatment. After an initial period of delay determined by the time required to

complete the transcription started prior to rifampicin addition (*Chen et al., 2015a*), the abundance of each individual mRNA species decays exponentially to a stable baseline (*Figure 4—figure supplement 1A*). From these decay curves, we extracted the half-life for each mRNA using a simple model for RNA decay (Materials and methods). Both the decay curves and the extracted half-lives were highly reproducible between biological replicates (*Figure 4—figure supplement 1A–D*).

To identify potential effects of localization on the lifetime of mRNAs, we sorted the measured half-lives of mRNAs into four groups based on the predicted locations of the encoded protein (cytoplasmic, periplasmic, inner-membrane and outer-membrane). Within each group, we observed significant variation between half-lives for individual mRNAs, as previously observed (*Bernstein et al., 2002*; *Chen et al., 2015a*; *Selinger et al., 2003*). Despite this spread, mRNAs that encode cytoplasmic, periplasmic, or outer-membrane proteins had half-lives that were similarly distributed and the lifetime distributions of these three groups were statistically indistinguishable according to a two-sided Kolmogrov-Smirnov test (*Figure 4A,B*). By contrast, mRNAs that encode inner-membrane proteins were degraded substantially more rapidly, on average, than the other three groups of mRNAs, exhibiting a statistically significantly different lifetime distribution (*Figure 4A,B*).

To determine if the native destabilization of the inner-membrane-protein mRNAs was related to the membrane localization of these mRNAs, we treated cells with kasugamycin to remove mRNAs from the membrane and repeated the τ-seq measurements. This treatment preferentially stabilized the mRNAs that encode inner-membrane proteins (*Figure 4—figure supplement 2*), and after the treatment the average lifetime of this group of mRNAs became comparable to those of the other three groups (*Figure 4C,D*). These results suggest that membrane localization is important to the native destabilization of these mRNAs.

## Artificially induced membrane localization destabilizes mRNAs

To further correlate mRNA lifetime with cellular localization without the global perturbation to cellular metabolism introduced with kasugamycin, we measured how the lifetimes of mRNAs were affected by fusion with signal-peptide sequences that target mRNAs to different cellular locations. However, it is known that the degradation rates of mRNAs depends on their sequences (*Mackie, 2012*); thus, sequence changes to the mRNAs caused by such fusions could lead to additional, sequence-dependent changes in lifetime, complicating the interpretation of such measurements. To overcome this challenge, we developed an approach to measure the lifetimes of a large number of fusion RNAs so that the average effect that arises from the cellular localization could be determined. Specifically, we exploited massively multiplexed cloning (*Kosuri and Church, 2014*) to create a large library of fusion constructs comprising ∼4800 distinct signal-peptide sequences fused to five different test mRNAs (mMaple3, neo, bla, lacZ, and phoA; *Figure 5A*). The ∼4800 signal-peptide sequences include the following groups: i) sequences of all 775 predicted *E. coli* SRP signal peptides; ii) sequences of all 431 predicted *E. coli* SecB signal peptides; iii) sequences from the N-terminal region of 400 cytoplasmic proteins; iv) synthetic, non-native encodings of i)–iii), which still encode the designated peptide sequences but with synonymous codons; and v) replicates of i)–iii) but with the first two codons, including the start codon, replaced with a pair of stop codons. Based on the results in *Figure 3*, we expect the group i) signal sequences to send the test mRNAs to the membrane, whereas group ii) signal sequences would not send the mRNAs to the membrane. Group iii) serves as an additional control that also should not send mRNAs to the membrane. We included group (iv) to test whether the effect on mRNA degradation rates is determined by the amino acid or mRNA sequence of the signal peptides and group (v) to test whether this effect requires translation.

We next used τ-seq to measure the lifetimes of these ∼24,000 fusion mRNAs, utilizing the signal peptide as a variable barcode to identify each of the different constructs (*Figure 5A*). For all five test genes, fusions to SRP signal-peptide sequences (*Figure 5B*, red) had substantially lower average mRNA lifetimes than fusions to either SecB signals or the cytoplasmic controls (*Figure 5B*, blue and cyan), and in all cases the SRP fusions were the only statistically distinguishable set of half-lives as determined by a two-sided Kolmogrov-Smirnov test. This difference was not abolished by replacing the native codons of these signal-peptide sequences with randomly selected synonymous codons (*Figure 5—figure supplement 1*). Notably, inhibition of translation by deletion of the start codon removed the stability differences between fusions to SRP signal sequences and fusions to SecB or cytoplasmic controls (*Figure 5C*). Taken together, these results support a model in which translation

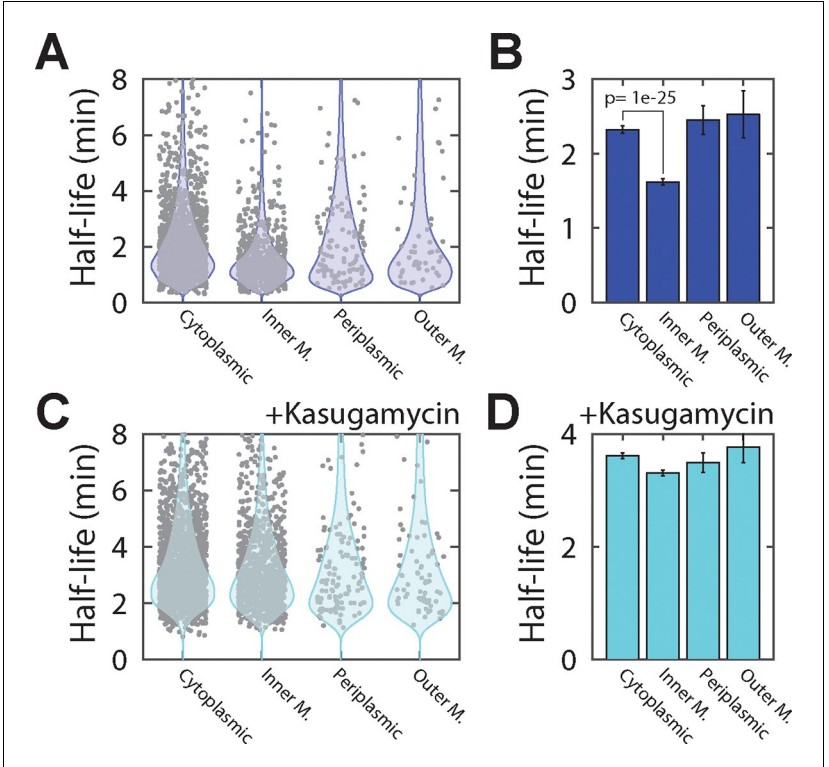

**Figure 4.** Inner-membrane-protein mRNAs are preferentially destabilized relative to mRNAs encoding cytoplasmic, periplasmic, and outer-membrane proteins. (**A**) Scatter plot (grey symbols) of the half-lives of individual *E. coli* mRNA species grouped based on the predicted locations of the proteins that they encode . Each data point represents one mRNA species. Blue colored shapes represent the probability distributions for these data points. (**B**) Average half-lives of the mRNA groups depicted in (**A**). The p-value was determined with a two-sided Kolmogrov-Smirnov test. (**C, D**) Same as (**A, B**) but for cells after treatment with kasugamycin. See *Figure 4—source data 1* for all abundance data versus time and the fit decay rates used to derive half-lives.

The following source data and figure supplements are available for figure 4:

**Source data 1.** RNA abundance measurements versus time and half-lives derived from these data for wild-type *E. coli* in the presence and absence of kasugamycin.

**Figure supplement 1.** Reproducibility of τ-seq measurements between biological replicates.

**Figure supplement 2.** The ratio of mRNA half-lives in the presence and absence of kasugamycin.

of SRP signal-peptide sequences that direct mRNAs to the membrane in a translation-dependent manner preferentially destabilize these mRNAs.

However, the observed destabilization in the SRP-fusions need not be entirely due to mRNA localization. For example, it has been previously recognized that the RNA sequences encoding signal peptides have a sequence bias (*Prilusky and Bibi, 2009*). Indeed, we find a correlation between the nucleotide content of the different fusion sequences and the lifetime of the fusion (*Figure 5—figure supplement 2*): nucleotide sequences encoding SRP-dependent signal peptides tend to be depleted of adenosine (A), and mRNA sequences with a lower A content tend to have a lower lifetime. This observation reveals that sequence bias of the RNA encoding the signal peptides also plays a role in the relative destabilization observed between the SRP fusions and all other fusions (*Figure 5B*) and raises the possibility that such sequence bias might explain a part of the destabilization observed for native mRNAs that encode inner-membrane proteins (*Figure 4A,B*). However, after controlling for this sequence bias in the signal peptide fusions, we still observed a relative destabilization of the

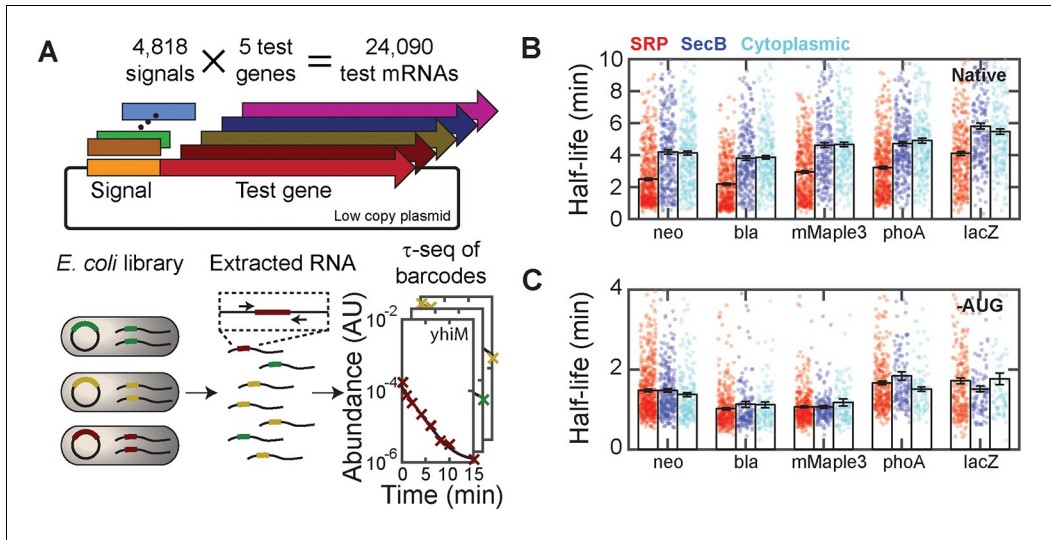

**Figure 5.** Targeting mRNAs to the membrane reduces their lifetimes. (**A**) Schematic diagram describing the construction of ~24,000 unique fusions of signal peptide sequences and test genes, and the measurement of the lifetime of the mRNA for each of these fusion constructs. Only the variable signal peptide region is amplified and sequenced; thus, it also serves as a unique barcode for each construct. (**B**) The mRNA half-lives of all fusion constructs between various signal peptides (SRP, red; SecB, blue; and cytoplasmic-control, cyan) and different test genes (neo, bla, mMaple3, phoA and lacZ). The mRNA lifetimes of fusion constructs with SRP signal peptides are statistically significantly different from those of the fusion constructs with SecB signal peptides or cytoplasmic controls, as determined by a two-sided Kolmogrov-Smirnov test. These p-values are $4\times10^{-20}$, $2\times10^{-22}$, $2\times10^{-15}$, $1\times10^{-21}$, and $3\times10^{-10}$ for difference between the SRP and SecB fusions for neo, bla, mMaple3, phoA, and lacZ, respectively. (**C**) As in (**B**) but for the fusion mRNAs in which the start codon is replaced by a stop codon (-AUG). Colored symbols in (**B**) and (**C**) represent lifetimes of individual mRNA species, and black bars represent the mean for each group. All error bars represent standard error of the mean. See *Figure 5—source data 1* for all abundance data versus time and the fit decay rates used to derive half-lives.

The following source data and figure supplements are available for figure 5:

**Source data 1.** RNA abundance measurements versus time and half-lives derived from these data for all signal-peptide fusions.

**Figure supplement 1.** Half-lives of fusion constructions between five test mRNAs and the native or synthetic encodings of various signal peptides.

**Figure supplement 2.** Effect of sequence bias on the half-lives of the SRP-fusion mRNAs.

SRP fusions as compared to other fusion groups for sequences containing the same number of A, T, G, or C nucleotides (*Figure 5—figure supplement 2*), supporting the model in which a portion of this destabilization is due to the membrane enrichment of these mRNAs as established by translation of the SRP-dependent signal peptide.

Interestingly, the two different modes of translation inhibition that we employed produced different global effects on mRNA lifetimes, with the average lifetime increased upon inhibition of ribosome assembly on mRNAs via kasugamycin treatment (*Figure 4D*) and decreased upon removal of the start codon (*Figure 5C*). These contrasting changes in mRNA stability suggest that the coupling between translation and degradation may be more complicated than previously anticipated (*Mackie, 2012*). Nonetheless, despite their differential effects on mRNA lifetimes, both treatments abrogated the stability differences between mRNAs encoding inner-membrane proteins and mRNAs encoding proteins in other cellular compartments, supporting the model in which the native destabilization of the inner-membrane-protein mRNAs arises from the translation-dependent membrane localization of these mRNAs.

It is also worth noting that the signal-peptide fusion experiments also revealed a surprising degree of variability in the rate at which mRNAs are degraded in *E. coli*. The nucleic acid sequences encoding these signal peptides account for no more than ~5% of the total mRNA sequences, yet, for each test gene and each group of signal peptides, variation in this portion of the mRNA can cause up to ten-fold differences in the lifetime (*Figure 5B*). Such a high sensitivity of lifetime to sequence suggests that the cell could fine tune lifetimes through modest changes to sequence.

Finally, it has been proposed that the nucleotide sequences encoding signal peptides as well as those flanking such regions have been evolutionarily optimized to introduce translational pauses that facilitate membrane targeting and co-translational insertion (*Fluman et al., 2014*). Specifically, it has been shown that sequences that cause translational pauses are enriched in regions flanking the sequences encoding the SRP-signal-peptide, and this observation has led to the proposal that such sequence-induced pauses may help improve membrane targeting and prevent the translation of cytotoxic membrane proteins in the cytoplasm (*Fluman et al., 2014*). However, we observe membrane targeting (*Figure 3*) and the corresponding decrease in half-life (*Figure 5* and *Figure 5—figure supplement 1*) in constructs that do not contain such pause sequences: the constructs to which we fuse the signal peptides are not membrane proteins and, thus, will not have these sequences, and one set of our SRP fusions uses synthetic encodings of the SRP sequences which would likely eliminate or weaken any nucleotide-sequence-based signals. Thus, our results indicate that such cis-acting nucleotide sequence features are not required for membrane localization; though, we cannot rule out the possibility that they improve the performance of SRP targeting.

## Membrane localization of RNA degradation enzymes preferentially destabilizes inner-membrane protein mRNAs

Next, we investigated the mechanism responsible for the preferential destabilization of mRNAs localized at the membrane. We reasoned that spatial organization of the mRNA processing and degradation enzymes might play a role in this effect since several RNA degradation enzymes have been found on the membrane in *E. coli* (*Mackie, 2012*). However, out of the roughly 20 enzymes involved in RNA processing and degradation, the localization of only a handful of these enzymes have been studied previously (*Mackie, 2012*). Thus, to understand the full extent of spatial organization in this pathway, we created C-terminal fusions between each of these proteins and the monomeric, photo-activatable fluorescent protein mMaple3 (*Wang et al., 2014*) at the native chromosomal locus and measured the distribution of these enzymes using 3D STORM in live cells. Of the 24 enzymes measured, we observed that only four — the endonuclease RNase E, the 3′-5′ exonuclease PNPase, the RNA helicase RhlB, and the poly-adenylation enzyme PAPI — were enriched at the membrane, whereas the remaining proteins were largely uniformly distributed throughout the cell (*Figure 6A,B* and *Figure 6—figure supplement 1*). These four enzymes are part of a multi-enzyme complex called the RNA degradosome (*Mackie, 2012*), which is known to bind to the membrane via a short amphipathic helix, segment A, that is internal to RNase E (*Khemici et al., 2008*). To confirm that all four enzymes are indeed enriched at the membrane due to the membrane anchoring of RNase E, we constructed a strain in which segment A of RNase E was removed (ΔA) and found that these enzymes were no longer localized at the membrane (*Figure 6C,D*).

To determine if the spatial organization of the RNA degradosome plays a role in the native destabilization of inner-membrane-protein mRNAs, we exploited the fact that deletion of segment A completely abrogates membrane localization of RNA degradation enzymes and repeated our τ-seq measurements in the ΔA strain. As expected from the sensitivity of the enzymatic activity of RNase E to lipid binding (*Murashko et al., 2012*), deletion of segment A led to a global stabilization of mRNAs. However, not all four groups of mRNAs were equally affected. Remarkably, this perturbation preferentially stabilized mRNAs (*Figure 6—figure supplement 2*) encoding inner-membrane mRNAs to the degree that the lifetimes of this group were no longer statistically distinct from the lifetimes of the other three groups (*Figure 6E,F*). In total, these observations favor a model in which the spatial proximity between the membrane bound RNA degradosome and membrane-localized mRNAs leads to a specific increase in the turnover rates of these mRNAs (*Figure 7*).

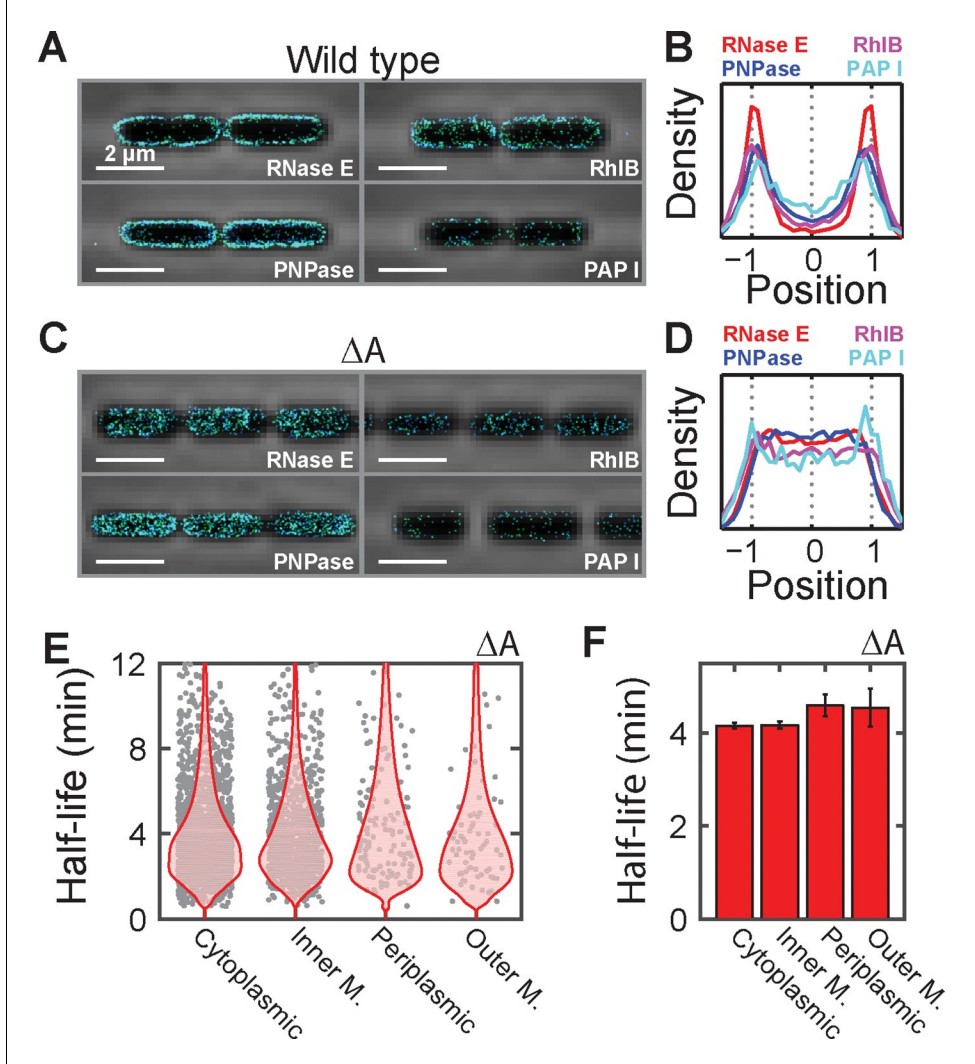

**Figure 6.** Membrane localization of RNA degradation enzymes is required for the preferential destabilization of inner-membrane-protein mRNAs. (**A**) Stacked phase contrast (gray) and STORM cross-section images (color) of example *E. coli* cells expressing mMaple3 fused to RNase E, RhlB, PNPase, and PAPI in the wild-type background. (**B**) Density profiles of RNase E, RhlB, PNPase, and PAPI in the wild-type background. Density profile is as defined in *Figure 1E*. (**C, D**) Same as (**A, B**) but for the ΔA strains where the membrane anchor of RNase E, segment A, is deleted. (**E**) Scatter plot (grey symbols) of half-lives of *E. coli* mRNAs in the ΔA strain grouped based on the predicted locations of the encoded proteins, shown together with the associated probability distributions (red). (**F**) Average half-lives for the mRNA groups depicted in (**E**). Error bars represent standard error of the mean. Scale bars: 2 μm. Average density profiles in B and D were derived from all measured cells, tens to hundreds of cells for each strain. See *Figure 6—source data 1* for all abundance data versus time and the fit decay rates used to derive half-lives.

The following source data and figure supplements are available for figure 6:

**Source data 1.** RNA abundance measurements versus time and half-lives derived from these data for the mutant *E. coli* strain.

**Figure supplement 1.** The spatial distribution of RNA processing enzymes in *E. coli*.

**Figure supplement 2.** The ratio of half-lives between a degradosome mutant and the wild-type strain.

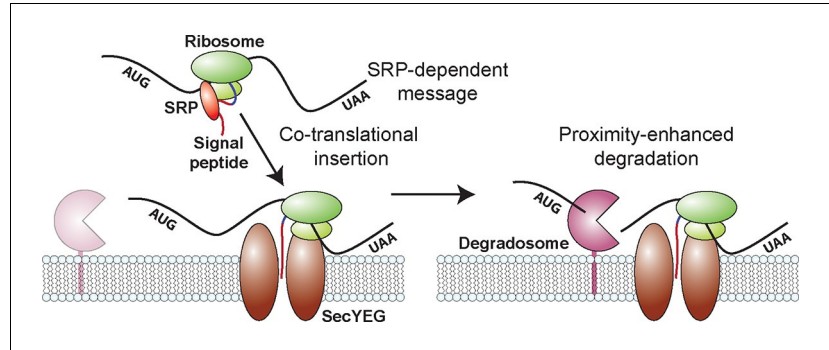

**Figure 7.** A model for the molecular mechanisms underlying the membrane localization of mRNAs encoding inner-membrane proteins and the role of this membrane localization in mRNA degradation. Translation of SRP-dependent signal peptides recruits SRP and directs mRNAs to the membrane, where the nascent polypeptide is co-translationally inserted in the membrane pore, SecYEG. Proximity of the membrane-bound RNA degradosome to these membrane-localized mRNAs leads to a preferential destabilization of these mRNAs.

## Discussion

In this work, we examined the spatial organization of the transcriptome of the model bacterium *E. coli* using a novel FISH-based RNA imaging approach that allowed us to measure the spatial organization of large and defined fractions of the transcriptome directly. These measurements revealed a transcriptome-scale spatial organization of the *E. coli* mRNAs: mRNAs that encode inner-membrane proteins are strongly enriched at the membrane while mRNAs that encode cytoplasmic, periplasmic and outer-membrane proteins are relatively diffusely distributed throughout the cytosol (*Figure 1*). In total, we imaged 75% of the expressed *E. coli* transcriptome; thus, we conclude that the distribution patterns observed here are the basal patterns of the spatial organization of mRNAs, although we cannot rule out the possibility that the distributions of some individual mRNAs may deviate from these transcriptome-wide patterns.

We also elucidated the molecular mechanism that gives rise to the observed spatial organization. Our experiments show that the membrane localization of mRNAs encoding inner-membrane proteins depends on translation and is most likely caused by co-translational insertion of the membrane proteins mediated by the SRP pathway (*Figure 3*). These results are consistent with previous biochemical studies showing that co-translational insertion is the dominant pathway for targeting these proteins to the bacterial membrane (*Driessen and Nouwen, 2008*) and are in keeping with the notion derived from previous nucleoid imaging studies that these membrane proteins can be translated and inserted into the membrane while their mRNAs are still being transcribed (*Bakshi et al., 2015*).

We further discovered that the spatial organization of the *E. coli* transcriptome has a physiological consequence on the post-transcriptional dynamics of mRNAs encoding inner-membrane proteins: these mRNAs are degraded more rapidly, on average, than mRNAs that encode cytoplasmic, periplasmic and outer-membrane proteins (*Figure 4*). Moreover, this native destabilization of the inner-membrane protein mRNAs depends on their membrane localization (*Figures 4* and *5*) and on the membrane localization of the RNA degradosome (*Figure 6*). Removal of this degradation machinery from the membrane preferentially stabilized inner-membrane protein mRNAs and equalized the stability of all four groups of mRNAs (*Figure 6*). Thus, our measurements suggest a model in which proximity between the membrane-bound degradation machinery and the mRNAs encoding inner-membrane-proteins, which are localized at the membrane by co-translational membrane insertion of the proteins, is at least in part responsible for the preferential destabilization of this group of mRNAs (*Figure 7*). It is possible that the transcriptome organization that we observe might also play a role in shaping the dynamics of other aspects of the life of a bacterial mRNA, e.g. transcription and translation. However, because our measurements do not discriminate between nascent versus fully formed RNA or RNAs that are or are not being translated, the data we present here provide little insight into these processes.

Finally, there are reasons to believe that the proximity-enhanced degradation mechanism that we discovered for *E. coli* may be found in a wide range of prokaryotes. Several studies have revealed that a surprisingly disparate set of prokaryotes anchor core components of their RNA degradation machinery to the membrane. Both the gram positive bacterium *B. subtilis* and the archaeon *S. solfataricus* have membrane-bound degradosomes (or exosomes in the case of archaea). Remarkably, each organism uses a unique localization mechanism that differs from that utilized by *E. coli* (*Lehnik-Habrink et al., 2011*; *Mackie, 2012*; *Roppelt et al., 2010*), suggesting that this organization is the result of convergent evolution. Moreover, the SRP-pathway is also broadly conserved in all forms of life. Based on these observations, we anticipate that the co-translational insertion of integral membrane proteins may lead to similar membrane localization of their mRNAs and therefore an enhanced turnover of these mRNAs in a wide range of bacterial organisms. Our results thus reveal that prokaryotes, like eukaryotes, can also use spatial organization to modulate the post-transcriptional fate of RNAs.

## Materials and methods

### Complex FISH probe design and construction

We generated our FISH probes using enzymatic amplification of array-based oligonucleotides libraries (*Beliveau et al., 2012*; *Chen et al., 2015b*; *Murgha et al., 2014*) (*Figure 1B*). Individual template molecules were designed by concatenating the following sequences: i) an index primer (I1) unique to the specific probe group, ii) the common reverse transcription primer (RTP) P9 (CAG GCA TCC GAG AGG TCT GG), iii) the site for the nicking enzyme Nb.BsmI, iv) the reverse complement of the targeting region (TR) designed to hybridize to a specific cellular RNA, v) the reverse complement of the nicking enzyme Nb.BsrDI, and vi) a second index priming site (I2) unique to the specific probe group. Targeting regions were designed using OligoArray 2.0 (*Rouillard et al., 2003*) and all annotated *E. coli* transcripts (K-12 mg1655; NC90013.2) with the following constraints: 30-nt length, a 80–85°C melting-temperature range for the duplex formed between the targeting region and the cellular RNA, a 50–60% GC content range, and a 75°C maximum melting temperature for secondary structure and cross-hybridization between different targeting regions. Index primers were designed by truncating the members of an existing library (*Xu et al., 2009*) of 240,000 oligonucleotides, each 25-nt long, to 20-nt length and selecting oligos for melting temperatures of 65–70°C, GC content of 50–60%, the absence of contiguous runs of 4 or more identical bases, the presence of a 3' GC clamp, i.e. 2–3 G/C within the final 5 nucleotides, and low homology (<12-nt homology) to other primers as well as the *E. coli* transcriptome, the T7 promoter (TAA TAC GAC TCA CTA TAG GG), and the common reverse transcription primer (P9) using BLAST+ (*Camacho et al., 2009*).

To test the role of protein localization, mRNAs were grouped into three abundance ranges (1/3–3 mRNAs/cell; 3–30 mRNAs/cell; and 30–300 mRNAs/cell) and six cellular locations as predicted by the pSortB 3.0 server (*Yu et al., 2010*) (http://www.psort.org/psortb/): cytoplasm, inner membrane, periplasm, outer membrane, extracellular, or unknown. The extracellular and unknown groups were not studied in this work. To test the role of polycistrons in RNA localization, mRNAs groups were further subdivided by whether or not a given message is polycistronic with a message encoding inner-membrane proteins. All designed targeting regions for mRNAs within each group were utilized to make probes for that group with the exception of mRNAs that encode cytoplasmic proteins. 1/3 of the possible targeting regions for each gene that encodes a cytoplasmic protein were selected at random to limit the number of probes required for these stains. To test the role of genome organization, targeting regions for mRNAs transcribed from every other 100-kb region of the genome, e.g. 100–200 kb, 300–400 kb, etc., and within the abundance ranges of 1/3–3 mRNAs/cell or 3–30 mRNAs/cell were used.

Multiple probe template sets were combined into large oligopools, and these pools were synthesized via CustomArray (http://customarrayinc.com/). Template sequences are provided in *Supplementary file 1*. Template subsets were amplified and labeled using the following procedure (*Figure 1B*). First each subset was selected and amplified with limited-cycle PCR. These templates were then amplified using in vitro transcription. The RNA products were then converted back into DNA with reverse transcription using a fluorescently labeled primer (Alexa647-P9). The template RNA was removed with alkaline hydrolysis, and the probes were column purified (Zymo Oligo Clean

and Concentrator; D4060) using a published protocol (*Chen et al., 2015b*). Antisense control probe sets (*Figure 1—figure supplement 1E*) were created by PCR amplifying the complex oligopool with index primers in which the T7 promoter was switched to index primer 1. Probes were then produced using an Alex647-labeled index primer 2 as the reverse transcription primer.

## FISH staining and 3D-STORM imaging of RNA

Overnight cultures of *E. coli* were diluted 1:200 in Lennox Luria Broth (LB) and grown at 32°C with shaking (250 rpm) to an optical density at 600 nm ($OD_{600}$) of 0.3. Cells were fixed, permeabilized, and stained as described previously (*Skinner et al., 2013*) utilizing either complex FISH probes targeting individual groups of RNA described above or single-molecule FISH (smFISH) probes against mMaple3. smFISH probes to mMaple3 were designed as described previously (*Skinner et al., 2013*). Kasugamycin-treated cells were harvested 15 min after the addition of kasugamycin (Sigma; K4013) to a final concentration of 1 mg/mL. Cells were affixed to the surface of custom imaging chambers coated with 0.1% v/v ployethyleneimine (Sigma; P3143) for 15 min at room temperature.

Samples were imaged on a home-built STORM microscope described elsewhere (*Huang et al., 2008*). Alexa647 was excited with a 657-nm laser and reactivated with a 405-nm laser. Laser powers at 657 nm and 405 nm were 100 mW and 1 mW at the microscope backport, respectively. Oblique-incidence illumination was used for all measurements. The sample was imaged with a 100×, 1.40 NA, UPlanSApo Ph3 oil immersion objective (Olympus) and an EM-CCD camera (Andor; iXon-897). Z calibration was performed by imaging Alexa-647-labeled antibodies affixed to a coverslip scanned along the optical axis with an objective positioner (Mad City Labs; Nano-F). These data were analyzed using the previously reported 3D STORM method (*Huang et al., 2008*) and the open source software zee-calibrator (http://zhuang.harvard.edu/software.html). Images were analyzed with the algorithm 3D-daoSTORM (*Babcock et al., 2012*) and rendered with custom software written in Matlab (https://github.com/ZhuangLab/matlab-storm). Phase contrast images were collected before and after each STORM image.

Individual cells were identified and internal coordinate systems constructed using the phase contrast images and a custom implementation of previous algorithms (*Guberman et al., 2008*; *Sliusarenko et al., 2011*). Cell boundaries were identified with sub-pixel resolution from the contour of constant intensity corresponding to the region of steepest descent in the phase image. The two regions of largest curvature in this boundary were identified as the cell poles, and a center line was created between these poles. The boundary of the cell was then divided into 100 regions of equal arc length, and corresponding regions on each side of the cell were connected to form cellular 'ribs'. Spurious or filamentous cells were eliminated from subsequent analysis by discarding cells whose cell boundary lengths along each side of the center line were not within 20% of each other and by discarding cells whose total areas were larger than 3 $\mu m^2$. (See *Source code 1*).

Single-molecule localizations were mapped to the coordinate system of each cell based on their relative position to the centerline and the closest cellular ribs. Cell-to-cell variations in width, length, and curvature were removed by normalizing this coordinate system by the length of the center line and the length of the individual ribs. This transformation effectively maps each cell to a cylinder of a common length and a common radius, where X, Y, and Z correspond to the position along the center line; the distance from the center line in the imaging plane; and the distance from the center line along the optical axis. Average short-axis cross-section images, such as that in *Figure 1D* (left), were rendered from all molecules from all cells with Y positions within the central 150-nm thick slice of the cell. Average long-axis cross-section images such as that in *Figure 1D* (right) were rendered from all molecules from all cells with X positions in the middle 80% range to remove molecules at poles. Cross-sectional density profiles in *Figure 1E,H*, *Figure 3D*, and *Figure 6B,D* were created from a histogram of all localizations along the normalized Y direction falling within the middle 50% of the normalized Z range and the middle 80% of normalized X range.

## τ-Seq measurements of endogenous mRNAs

Cells were harvested as a function of time after rifampicin addition from *E. coli* cultures grown as described above to an $OD_{600}$ of 0.4 using a previously published protocol (*Bernstein et al., 2002*). In vitrotranscribed RNAs (spike-ins) were added for normalization between time points, and total RNA was harvested using the RNAsnap protocol (*Stead et al., 2012*). The sequences of the spike-in

RNAs are available upon request. DNase I was used to remove genomic contamination, and rRNA was removed using the Gram-Negative RiboZero kit (Epicentre; MRZGN126). Sequencing libraries were constructed using the RNA Ultra Directional Kit (New England Biolabs; E7420). 50-bp or 75-bp single-ended sequencing of τ-seq samples was performed on either the Illumina HiSeq2000 or the NextSeq500. All sequencing data are available via GEO accession GSE75818.

Sequencing data were aligned to the mg1655 genome (NC_000913.2) using bowtie 0.12.9 (*Langmead et al., 2009*). Reported counts per mRNA were determined by summing counts corresponds to the region between the start and stop codons of each gene. The abundance of the in vitro spike-ins in combination with the published conversion between $OD_{600}$ and cell number (*Volkmer and Heinemann, 2011*) were used to initially calibrate absolute copy numbers per cell. Using this calibration, it was determined that the stable RNA species, tmRNA, had an average copy number of 597 ± 27 (STD across the 8 time points from the first replicate of the wild-type strain in the absence of kasugamycin). The final calibration was performed by fixing the tmRNA concentration at all time points to this value, thereby eliminating small variations in RNA extraction efficiency between samples.

All decay profiles were fit with an expression that incorporates three features: i) a delayed onset of the exponential decay; ii) a period of exponential decay; and iii) a stable baseline. The delayed onset of decay arises because rifampicin is an initiation inhibitor not an elongation inhibitor; thus, there is a period of time during which RNAs continue to be transcribed (*Chen et al., 2015a*). During this period of time, transcription continues to replenish degraded RNAs and the system is effectively at steady-state. Thus, we fit the number of RNA molecules as a function of time, *N(t)*, with the following piecewise function:

$$N(t) = N_f + N_0 \begin{cases} 1 & t \le \alpha \\ e^{-k(t-\alpha)} & t > \alpha \end{cases} \tag{1}$$

where $N_0 + N_f$ is the initial number of mRNA molecules, $N_f$, is the number of mRNA molecules in the stable baseline, $k$ is the rate of exponential decay, and $\alpha$ is the duration of the initial delay before net decay begins. Conceptually, $\alpha$ is related to the time required for the last round of polymerases bound prior to the rifampicin treatment to complete synthesis of the given gene. The duration of this delay depends linearly on the distance between the specific portion of an mRNA being measured and the promoter; thus, messages that are at the end of polycistronic mRNAs will have a larger $\alpha$ value than messages that are at the beginning of polycistronic messages or are not members of polycistronic messages (*Chen et al., 2015a*). Our observations indeed confirmed this prediction. *Equation (1)* can be viewed as an approximation for more complicated models that restrict when RNA degradation can begin, i.e. co- or post-transcriptionally, or incorporate the finite time required for RNAP polymerase to transcribe a message of a given length (*Chen et al., 2015a*). Such additional complications soften the boundary between the constant and exponential decay phases by introducing additional piece-wise components that contain linear or quadratic decays (*Chen et al., 2015a*). A non-linear least squares algorithm was used to fit the natural logarithm of *Equation (1)* to the natural logarithm of the τ-seq decay profiles. This logarithmic transformation equalized the weighting of all abundance measurements in the fitting routine. Reported half-lives are determined from the fit decay rates via $\tau = \log(2)/k$.

Half-lives are reported only if the error (estimated as 1/4 of the 95% confidence interval of this value returned by the fit) of the corresponding decay rate is less than half of the measured decay rate. Where applicable the reported half-lives are the average across two biological replicates. Measured half-lives larger than our final time point (20 min) were also excluded because *Equation (1)* was unreliable in fitting such decay curves given the time resolution of our measurement.

## Barcoded τ-seq measurements of mRNAs of the signal-peptide fusion libraries

The signal-peptides in these libraries were designed by submitting all gene sequences from the annotated mg1655 genome (NC_000913.2) to the following servers: pSortB 3.0 (*Yu et al., 2010*) (http://www.psort.org/psortb/), signalP 4.0 (*Petersen et al., 2011*) (http://www.cbs.dtu.dk/services/SignalP/) and TMHMM 2.0 (*Krogh et al., 2001*) (http://www.cbs.dtu.dk/services/TMHMM/). SRP-dependent proteins were defined as proteins with more than one TMHMM-predicted transmembrane domain and which were predicted to reside within the inner-membrane by pSortB. The signal

peptide was derived from a 30-amino-acid region centered on the first TM domain. If this region exceeded the N-terminus of the protein, the first 30 amino acids at the N-terminus of the protein were used as the signal peptide. SecB-dependent proteins were defined as proteins predicted to contain a SecB-dependent signal peptide via the signalP server. Because of the similarity between N-terminal transmembrane domains and SecB-dependent signal sequences (both are highly hydrophobic), some N-terminal transmembrane domains that are identified as SecB signal peptides via signalP are also identified as transmembrane domains via TMHMM, and are thus likely SRP-dependent proteins. To eliminate these spurious SecB signals, we removed predicted SecB signals derived from proteins that TMHMM predicted to have two or more transmembrane domains. The non-native SecB-dependent protein, beta lactamase, was also included in this set. The cytoplasmic control proteins were selected at random from proteins predicted to reside in the cytosol by pSortB and which were not included in either of the SRP or SecB groups. The first 30 amino acids of the SecB-dependent and cytosolic control proteins were used as the signal peptide sequence. Three encodings were used for each signal sequence: i) the native *E. coli* encoding, ii) a synthetic encoding, and iii) an untranslated encoding. The synthetic encoding was generated by replacing each of the 30 codons in the native encoding by randomly selected synonymous codons using the codon usage across the *E. coli* genome as relative selection weights. The untranslated encoding was generated by replacing the first two codons of the native *E. coli* encoding with a pair of TAA stop codons.

A complex oligopool containing the desired signal peptide sequences was synthesized by CustomArray, amplified via PCR, and inserted via Gibson assembly (*Gibson et al., 2009*) into pZ-series plasmids (*Lutz and Bujard, 1997*) containing the desired genes. All signal peptides were linked to the test proteins via a common flexible linker, GGSGGS. The sequences for the signal peptides are provided in *Supplementary file 2*. RNA samples were prepared as described in the "τ-Seq measurements of endogenous mRNAs" section, albeit with a different set of in vitro spike-in molecules. Sequences are available upon request. cDNA was constructed for the signal peptide region only and amplified using a mixture of primers targeting the common flexible linker and differing only in the length of a stretch of random nucleotides. This stretch of random nucleotides introduced a random length offset that was required to overcome sequencing challenges with the NextSeq500 due to regions of low complexity in these libraries. The cDNA was amplified and sequenced, and relative abundances for each library member were determined using the abundance of the spike-in molecules. The resulting decay curves were fit with an exponential decay to a stable baseline. All sequencing data are available via GEO accession GSE75818.

## Cloning

All plasmids were created with Gibson assembly (*Gibson et al., 2009*) and are based on the pZ-series plasmids (*Lutz and Bujard, 1997*). Chromosomal integrations were created using the lambda red recombination system (*Datta et al., 2006*). All plasmids and strains reported here are summarized in *Supplementary file 3* and are available upon request.

## 3D-STORM imaging of RNA degradation proteins in live cells

Overnight cultures were diluted 1:10000 into MOPS minimal defined media supplemented with 0.2% w/v glucose and 34 μg/mL chloramphenicol and grown at 32°C to an $OD_{600}$ of 0.2. This medium was used to reduce autofluorescence in imaging measurements. Cells were concentrated thirty-fold and spotted onto sub-micron patterned agarose pads containing grooves to control cell density and orient cells (*Moffitt et al., 2012*). Cells were imaged at room temperature on a home-built microscope and published protocols (*Huang et al., 2008*; *Wang et al., 2011*). Briefly, mMaple3 was excited with a 561-nm laser and activated with a 405-nm laser, utilizing 100 mW and 1 mW at the microscope backport, respectively. Z-calibration was conducted with antibodies conjugated to Cy3. STORM Z-calibration, image reconstruction, and image rendering were conducted as described in the "FISH-staining and STORM imaging of RNA" section above.

## Acknowledgements

We thank H Babcock, SH Shim, G Dempsey for advice on instrumentation; and H Chen, K Shiroguchi, and C Daly for advice regarding sequencing. This work was in part supported by NIH grant R01GM096450 to XZ JRM was a Helen Hay Whitney postdoctoral fellow. ANB is a Damon Runyon

postdoctoral fellow. SW is a Jane Coffins Child postdoctoral fellow. XZ is a Howard Hughes Medical Institute investigator.

## Additional information

### Competing interests

XZ: Reviewing editor, *eLife*. The other authors declare that no competing interests exist.

### Funding

| Funder | Grant reference number | Author |
|---|---|---|
| National Institute of General Medical Sciences | GM096450 | Xiaowei Zhuang |
| Howard Hughes Medical Institute | HHMI | Xiaowei Zhuang |

The funders had no role in study design, data collection and interpretation, or the decision to submit the work for publication.

### Author contributions

JRM, Conception and design, Acquisition of data, Analysis and interpretation of data, Drafting or revising the article, Contributed unpublished essential data or reagents; SP, ANB, SW, Drafting or revising the article, Contributed unpublished essential data or reagents; XZ, Conception and design, Analysis and interpretation of data, Drafting or revising the article

### Author ORCIDs

Xiaowei Zhuang, http://orcid.org/0000-0002-6034-7853

## Additional files

### Supplementary files

• Supplementary file 1. Template sequences of the complex FISH probe sets. 'Library name' specifies whether the probes are for testing the role of the proteome organization or the genome organization. 'Experiment number' is a unique number for each experiment. 'Location' specifies either the location of the encoded protein or the location of the genomic locus. 'Abundance' specifies the abundance range for the targeted RNA in copies/cell. 'Polycistronic with inner membrane proteins' specifies whether the gene is a member of a polycistronic mRNA with a gene that encodes an inner-membrane protein (Y) or not (N). 'Gene name' is the name of the gene. 'Probe location' is the location of the 5' end of the target region in nucleotides from the start codon of the specified gene. 'Template sequence' is the sequence of the oligonucleotide used as a template for that FISH probe, which concatenates the index primers, the reverse transcription primer, and the targeting sequence in the order depicted in *Figure 1*. 'Index primer 1' and 'Index primer 2' are the sequences of the primers used to amplify the subset of probes that includes the specified probe.

• Supplementary file 2. Sequences of the N-terminal fusion constructs used to create the large-scale signal-peptide libraries. 'Gene name' is the name of the gene from which the signal peptide was derived. 'Signal-peptide type' specifies whether the signal peptide is a SRP, SecB, or cytoplasmic control peptide. 'Encoding' specifies the type of the encoding of the signal peptide. Native indicates that the nucleotide sequence is the native *E. coli* sequence; Synthetic indicates that the native *E. coli* codons have been exchanged at random with synonymous codons; and 'No translation' indicates that the first two codons, including the start codon, of the native *E. coli* sequence have been replaced with stop codons. 'Oligo sequence' specifies the sequence of the oligonucleotide used to create the specific fusion.

• Supplementary file 3. Plasmids and *E. coli* strains used in this work. 'Strain' is the name of the strain or strain library. 'Description' is a brief description of the strain and its purpose in this study. 'Parent

strain' lists the strain used to generate each strain where appropriate. 'Genotype' provides a compact description of the modifications to the strain. 'Plasmid' lists the name of the plasmids contained by each strain where appropriate. libJMXX represents a complex plasmid library. pZS*32 follows the naming convention of the pZ plasmids (Lutz and Bujard, 1997). SCS31 represents a selection-counter-selection cassette employing pRhaB-ccdB and the chloramphenicol resistance gene cat.

• Source code 1. Matlab functions for the identification of bacterial cell boundaries and the rendering of STORM images.

### Major datasets

The following dataset was generated:

| Author(s) | Year | Dataset title | Dataset URL | Database, license, and accessibility information |
|---|---|---|---|---|
| Moffitt JR, Zhuang XZ | 2015 | RNA life time measurements in E. coli | http://www.ncbi.nlm.nih.gov/geo/query/acc.cgi?acc=GSE75818 | Publicly available at Gene Expression Omnibus (accession no: GSE75818) |

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
