## [Decision Letter]

Thank you for submitting your work entitled "Spatial Organization Shapes the Turnover of a Bacterial Transcriptome" for consideration by *eLife*. Your article has been reviewed by three peer reviewers, and the evaluation has been overseen by a Reviewing Editor and Naama Barkai as the Senior Editor.

The following individuals involved in review of your submission have agreed to reveal their identity: Xavier Darzacq, Calvin Jan and Arjun Raj (peer reviewer).

The reviewers have discussed the reviews with one another and the Reviewing Editor has drafted this decision to help you prepare a revised submission.

Summary:

We have received three expert reviews on your manuscript submitted to *eLife*, all of whom find the work to be exciting and well executed and the manuscript clearly written. The insights revealed about mRNA localization in *E. coli* are compelling – that only the inner membrane protein mRNAs are localized to the membrane, and that their relatively rapid turnover is triggered by membrane-localized degradasome components. The data arguing that translation of the signal peptide, independent of codon, is critical to the mRNA localization event – as distinct from what occurs in eukaryotes – were also compelling. The reviewers had some modest concerns about the discussion of the coupling of transcription with translation that they thought could be addressed somewhat more clearly in the Discussion section. Additionally, the reviewers wondered whether the authors could speculate about the functional consequences of the shortened mRNA half-lives of the inner membrane protein mRNAs.

*Reviewer #1:*

In their manuscript "Spatial Organization Shapes the Turnover of a Bacterial Transcriptome" Jeffrey Moffitt and coworkers address the general distribution of mRNAs in the bacterium *E. coli*. mRNA localization in bacteria has been very confusing, while it is clear that nascent mRNAs can be co-transcriptionally translated and therefore transcription and translation are coupled, data provided by this study shows that most mRNAs are present outside the bacterial nucleoid. While this part of the work is very interesting I believe the manuscript in its current form lacks a clear statement describing the possibility that the presented data does not address the possibility that mRNAs are translated at least once co transcriptionally. I strongly believe that this aspect should be discussed. In a second part the main finding of the paper is a clear identification and exploration of SRP mediated mRNA localization to the periphery of the cells. Results are convincing and clear and the methodology is sound from a well-established lab. It is of general interest to have a global description of mRNA localization. The impact of mRNA SRP anchoring on mRNA stability and its link to the degradation machinery localization is of general interest.

As a general comment, while the study uses large numbers of mRNAs it does not provide a single mRNA specific view of RNA localization and therefore potential exceptions to the general rules or regulation mechanisms are not addressed. Finding that membrane localized mRNAs are less stable is interesting but lacks some functional assay. Since the authors delocalized the degradation machinery and stabilized the SRP population of mRNAs respective to the rest of the population, it would have been interesting to explore potential phenotypes (why exported proteins are degraded faster – is it just a peculiarity of this system or is it important for the bacteria?).

Overall I support publication in *eLife*, I recommend a clarification regarding co-transcriptional translation. A discussion on potential roles of mRNA destabilization at the membrane would be important and experiments monitoring phenotypes upon mRNA stabilization would be important to.

*Reviewer #1 (Additional data files and statistical comments):*

Data files of mRNA localisation would be useful.

*Reviewer #2:*

In their investigation of the spatial organization of the *E. coli* transcriptome, Moffitt et al. address a fundamental cell biological question that is well studied in eukaryotes but relatively poorly explored in bacteria. This manuscript advances our understanding of the cellular organization of mRNA metabolism in bacteria and merits publication in *eLife*.

The authors apply their expertise in super resolution microscopy and FISH to survey broad classes of mRNAs grouped by features that might influence their spatial organization in *E. coli*. They find that mRNAs encoding inner membrane proteins are enriched at the membrane, whereas essentially all other classes examined are distributed throughout the cytosol. The authors show that inner membrane localization depends on translation and requires a signal sequence, implicating SRP in the process. The resulting membrane localization increases susceptibility of these mRNAs to degradation by membrane-associated ribonucleases. Analysis of reporters bearing signal sequences demonstrates that this feature is sufficient to destabilize mRNAs and not simply a correlate of other cis-acting features or biases in nucleotide frequency.

To explore why membrane-associated mRNAs are generally less stable, the authors perform a systematic survey of RNase localization using C-terminal mMaple3-tagging for 3D STORM. The inner-membrane localization of RNaseE and other degradosome components was previously demonstrated, and the authors confirm these findings and further show that indeed they are the only membrane-enriched nucleases in *E. coli*. RNase E has a prominent role in mRNA turnover by catalyzing the primary endonucleolytic cleavage event that drives decay. RNase E can function in two modes, a 5' monophosphate-dependent recognition mechanism and a "direct entry" mechanism. Interestingly, the pyrophosphate hydrolase enzyme RppH is thought to be rate-limiting for RNA turnover in the 5' monophosphate-dependent recognition mode and is found by the authors to be cytosolic. It would be interesting to investigate the role of RppH in determining the stability of mRNAs encoding inner membrane proteins. No new experiments need to necessarily be performed, but the results merit discussion given the logic used throughout that localization influences function. Moreover, microarray analyses that compare steady-state mRNA abundance of wildtype and catalytically dead RppH strains could be re-analyzed in light of the encoded protein's localization (see Deana et al. doi:10.1038/nature06475). If mRNAs encoding inner membrane proteins are not stabilized by RppH depletion, then the high local concentration of RNaseE may favor their degradation by the direct entry pathway.

The main result from this study suggests that, at steady state, SRP is facilitating membrane targeting of mRNAs in bacteria. This result is worth highlighting given that SRP is not known to pause translation. Flurman et al. (doi:10.7554/*eLife*.03440) propose that programmed translational pauses, e.g. through SD/aSD interactions, promote membrane targeting in the absence of SRP-mediated pausing. The author's results are consistent with this model but do not dig into the mechanisms responsible for membrane localization. Given that the authors partially address the localization mechanism by ruling out cis-acting zipcodes, at least for the bglF mRNA, a broader discussion of how translation and localization are coupled in bacteria is encouraged. This is particularly relevant in the context of the heterologous reporters (Figure 5) for which there has been no selective pressure to optimize pausing, yet the mRNA stability data imply efficient membrane localization in the presence of a signal sequence. With respect to the bglF data, there are insufficient experimental details provided to determine if the mMaple3 reporters alter the 3´ UTR that may contain potential cis-acting sequences. This is an important detail that should be clarified.

*Reviewer #2 (Additional data files and statistical comments):*

The number of transcripts probed in each FISH set in each species group should be made more easily accessible. All sequencing data should be deposited at GEO or equivalent repository and the accession numbers added to the manuscript before publication.

*Reviewer #3:*

In this paper, Moffitt et al. develop a technique for looking at the spatial localization of the transcriptome. Using this method, they show that a class of mRNA (those encoding inner-membrane-proteins) preferentially localize to the periplasmic space. They then work out the mechanism by which these proteins localize to the periplasm, and show that this localization pattern causes these mRNA to degrade more quickly because of colocalization with various RNA degradation proteins.

I think this is a really nice piece of work. On a technical level, the images are stunning and, furthermore, very convincing. The idea of labeling classes of mRNA simultaneously is very clever and, even more impressive, the authors effectively use this method to explore some new biology. I think this work will have a real impact-there is a lot of interest in spatial organization in cells these days, but little solid evidence, and this paper is a big contribution in that respect.

The science, writing and presentation are all very solid. I also think the authors were very careful to craft their claims based on what their data show, which is commendable.

---

## [Author Response]

*Reviewer #1: In their manuscript "Spatial Organization Shapes the Turnover of a Bacterial Transcriptome" Jeffrey Moffitt and coworkers address the general distribution of mRNAs in the bacterium E. coli. mRNA localization in bacteria has been very confusing, while it is clear that nascent mRNAs can be co-transcriptionally translated and therefore transcription and translation are coupled, data provided by this study shows that most mRNAs are present outside the bacterial nucleoid. While this part of the work is very interesting I believe the manuscript in its current form lacks a clear statement describing the possibility that the presented data does not address the possibility that mRNAs are translated at least once co transcriptionally. I strongly believe that this aspect should be discussed.*

We agree with the reviewer that our data do not rule out the possibility that translation begins during transcription of many if not all mRNAs. We observe only a static snapshot of the distribution of mRNAs throughout the cell. Even in this snapshot, some mRNAs are observed inside the nucleoid. Moreover, for those mRNAs that were observed to be outside the nucleoid in the snapshot, our data do not rule out the possibility that these mRNAs have already been translated at least once while they were still inside the nucleoid. Finally, several studies have now shown that the translation machinery in *E. coli* is in part spatially organized (Bakshi et al., 2015; Robinow and Kellenberger, 1994), with translating ribosomes largely excluded from the nucleoid (Sanamrad et al., 2014). Based on these observations, our results do not rule out the possibility of co-transcriptional translation at all. We have added a couple of sentences in our revised Discussion section to state that our data provided little insights into the relation between transcription and translation (Discussion, end of third paragraph).

*In a second part the main finding of the paper is a clear identification and exploration of SRP mediated mRNA localization to the periphery of the cells. Results are convincing and clear and the methodology is sound from a well-established lab. It is of general interest to have a global description of mRNA localization. The impact of mRNA SRP anchoring on mRNA stability and its link to the degradation machinery localization is of general interest.*

*As a general comment, while the study uses large numbers of mRNAs it does not provide a single mRNA specific view of RNA localization and therefore potential exceptions to the general rules or regulation mechanisms are not addressed. Finding that membrane localized mRNAs are less stable is interesting but lacks some functional assay. Since the authors delocalized the degradation machinery and stabilized the SRP population of mRNAs respective to the rest of the population, it would have been interesting to explore potential phenotypes (why exported proteins are degraded faster – is it just a peculiarity of this system or is it important for the bacteria?).* We agree with the reviewer that our data provide some general principles for RNA localization in bacteria, and we do not rule out the possibility of potential exceptions, i.e. some specific RNAs may not follow the general rules presented in this paper. We explicitly state this in the manuscript (subsection “Spatial organization of the genome does not play a major role in the spatial organization of the *E. coli* Transcriptome”, last paragraph; Discussion, first paragraph).

We also agree that it would be interesting to explore the physiological consequences associated with perturbing the spatial organization of the degradosome. We have performed experiments along this line and the results show that perturbation to the membrane localization of the degradosome induces a significant growth cost (data not shown). A similar phenotype has been previously observed by others as well. Although it is tantalizing to attribute this growth phenotype in part to the delocalization of the degradosome, the interpretation of these data is, however, complicated because the enzymatic activity of RNase E also depends on its membrane attachment. Deletion of an amphipathic helix (segment A), removing RNase from the membrane, is known to reduce the catalytic activity of RNase E and leads to a global stabilizing effect on all mRNAs in addition to the preferential stabilization effect on inner-membrane-protein-encoding mRNAs due to delocalization of the degradosome. Hence the growth phenotype cannot be solely attributed to the perturbation of the membrane localization of RNase E. For this and other reasons, we believe that confirming that any phenotypes induced by this perturbation are indeed due to defects in mRNA localization will require a level of detailed experimentation beyond the scope of the current study.

*Overall I support publication in eLife, I recommend a clarification regarding co-transcriptional translation. A discussion on potential roles of mRNA destabilization at the membrane would be important and experiments monitoring phenotypes upon mRNA stabilization would be important to.*

We have addressed these points in the detailed responses above.

*Reviewer #1 (Additional data files and statistical comments): Data files of mRNA localisation would be useful.* Unfortunately, the size of these files prohibits us from providing them.

*Reviewer #2: In their investigation of the spatial organization of the E. coli transcriptome, Moffitt et al. address a fundamental cell biological question that is well studied in eukaryotes but relatively poorly explored in bacteria. This manuscript advances our understanding of the cellular organization of mRNA metabolism in bacteria and merits publication in eLife. […] To explore why membrane-associated mRNAs are generally less stable, the authors perform a systematic survey of RNase localization using C-terminal mMaple3-tagging for 3D STORM. The inner-membrane localization of RNaseE and other degradosome components was previously demonstrated, and the authors confirm these findings and further show that indeed they are the only membrane-enriched nucleases in E. coli. RNase E has a prominent role in mRNA turnover by catalyzing the primary endonucleolytic cleavage event that drives decay. RNase E can function in two modes, a 5' monophosphate-dependent recognition mechanism and a "direct entry" mechanism. Interestingly, the pyrophosphate hydrolase enzyme RppH is thought to be rate-limiting for RNA turnover in the 5' monophosphate-dependent recognition mode and is found by the authors to be cytosolic. It would be interesting to investigate the role of RppH in determining the stability of mRNAs encoding inner membrane proteins. No new experiments need to necessarily be performed, but the results merit discussion given the logic used throughout that localization influences function. Moreover, microarray analyses that compare steady-state mRNA abundance of wildtype and catalytically dead RppH strains could be re-analyzed in light of the encoded protein's localization (see Deana et al. doi:10.1038/nature06475). If mRNAs encoding inner membrane proteins are not stabilized by RppH depletion, then the high local concentration of RNaseE may favor their degradation by the direct entry pathway.*

The reviewer raised an interesting point: the lack of membrane localization for RppH and the idea that RppH catalyzed decapping of mRNA at the 5’ end is ratelimiting for RNA turnover in the 5' monophosphate-dependent recognition mode might suggest that inner-membrane-protein-encoding mRNAs are primarily degraded by the direct entry mode. Following the reviewer’s recommendation, we performed a simple analysis of the published mRNA abundance data in the WT and RppH mutant strains. Interestingly, we find that the abundance of mRNAs that encode inner-membrane proteins appear to be less affected by deletion of RppH than other mRNAs. This observation is indeed consistent with the notion that mRNAs that encode inner-membrane proteins are preferentially directed to the ‘direct entry’ pathway (RppH-independent pathway), and one might speculate that the proximity between these mRNAs and RNase E is the mechanism that favors this pathway.

Although tantalizing, we have decided not to include these results in our revised manuscript because, in our experience, changes in the mRNA abundance under perturbation to the RNA degradation machinery are imperfect predictors of the change in degradation rate because the abundance is determined not only by the degradation rate but also by the transcription rate, which could be indirectly affected by perturbations to the RNA degradation machinery. For this reason, direct measurements of the degradation rate in RppH mutants would be needed to conclusively address the reviewer’s question. These exciting experiments are planned for future studies.

*The main result from this study suggests that, at steady state, SRP is facilitating membrane targeting of mRNAs in bacteria. This result is worth highlighting given that SRP is not known to pause translation. Flurman et al. (doi:10.7554/eLife.03440) propose that programmed translational pauses, e.g. through SD/aSD interactions, promote membrane targeting in the absence of SRP-mediated pausing. The author's results are consistent with this model but do not dig into the mechanisms responsible for membrane localization. Given that the authors partially address the localization mechanism by ruling out cis-acting zipcodes, at least for the bglF mRNA, a broader discussion of how translation and localization are coupled in bacteria is encouraged. This is particularly relevant in the context of the heterologous reporters (Figure 5) for which there has been no selective pressure to optimize pausing, yet the mRNA stability data imply efficient membrane localization in the presence of a signal sequence.*

We agree that the relation between translational pausing and mRNA localization is an interesting subject. Unfortunately, our data add very little to this discussion because our images tell us nothing about translational pausing.

The reviewer raises the interesting point that our heterologous transcripts continue to display membrane targeting despite the fact that many of them (those with scrambled codon usage, in particular) do not contain nucleotide sequences that have been evolutionarily selected to optimize pausing. This result implies that such cis-acting sequences are not required for membrane targeting, and given the report of Fluman et al., we agree that this is a good point to make. We have now clarified this point in our revised text (See subsection “Artificially induced membrane localization destabilizes mRNAs“, last paragraph).

*With respect to the bglF data, there are insufficient experimental details provided to determine if the mMaple3 reporters alter the 3´ UTR that may contain potential cis-acting sequences. This is an important detail that should be clarified.* We thank the reviewer for raising this point. We have now clarified the exact sequence of these constructs and the specific model that they were designed to test in the revised main text (subsection “Co-translational insertion of membrane proteins is important for the membrane enrichment of the mRNAs encoding these proteins”, last paragraph). We performed these experiments to test the model proposed by Nevo-Dinur et al. 2011, which is that the membrane targeting of bglF mRNA is translation independent and that this targeting requires only a cis-acting RNA region that is within the coding region. The 3’UTR is not required in this model, rather, this proposed cis-acting region overlaps with the nucleotide sequence that encodes the signal peptide of bglF. Our constructs were designed to specifically test the model of Nevo-Dinur et al. and the role of translation of this putative cis-acting RNA region in mRNA localization. Thus, neither our constructs nor many of the constructs used in Nevo-Dinur et al. 2011 contained the native 3’UTR region. We have now clarified these points in the revised manuscript.

*Reviewer #2 (Additional data files and statistical comments): The number of transcripts probed in each FISH set in each species group should be made more easily accessible. All sequencing data should be deposited at GEO or equivalent repository and the accession numbers added to the manuscript before publication.* As suggested, we have included the number of transcripts probed in each FISH set in the captions of Figure 1—figure supplement 1. In addition, we have included the GEO accession number associated with our sequencing data (subsection “τ-Seq measurements of endogenous mRNAs“, first paragraph and subsection “Barcoded τ-seq measurements of mRNAs of the signal-peptide fusion libraries“, last paragraph).